# Bootstrapping Audio-Visual Segmentation by Strengthening Audio Cues

## Abstract

How to effectively interact audio with vision has attracted wide interest in the multi-modality community. Recently, a novel audio-visual segmentation (AVS) task has been proposed, aiming to segment the sounding objects in video frames using audio cues. However, current AVS methods suffer a modality imbalance issue. The fusion of audio features is insufficient because of its unidirectional and deficient interaction, while the vision information is more sufficiently exploited. Thus, the output features are always dominated by visual representations, which restricts audio-vision representation learning and may cause some false alarms. To address this issue, we propose AVSAC, where a Bidirectional Audio-Visual Decoder (BAVD) is devised with multiple bidirectional bridges built within. This strengthens audio cues and enables continuous interaction between audio and visual representations, which shrinks modality imbalance and boosts audio-visual representation learning. Furthermore, we introduce Audio Feature Reconstruction (AFR) to evade harmful data bias and curtail audio information loss by reconstructing lost ones from visual signals. Extensive experiments show that our method achieves new state-of-the-art performances in the AVS benchmark, especially boosting significant improvements (about 6% in mIoU and 4% in F-score) in the most challenging MS3 subset which needs to segment multiple sound sources.

## 1 Introduction

Audio and vision are two closely intertwined modalities that play an indispensable role in our perception of the real world. Recently, a novel audio-visual segmentation (AVS) (Zhou et al., 2022) task has been proposed, which aims at segmenting the sounding objects from video frames corresponding to a given audio. It consists of two sub-tasks: single sound source segmentation (S4) and multiple sound source segmentation (MS3). The task has rapidly raised wide interest from many researchers since being proposed, while there are still some issues remaining to be solved. One of the biggest challenges is that this task requires both accurate location of the audible frames and precise delineation of the shapes of the sounding objects (Zhou et al., 2022; 2023). This necessitates reasoning and adequate interaction between audio and vision modalities for a deeper understanding of the scenarios, which renders many present audio-visual task-related methods (Qian et al., 2020; Chen et al., 2021) unsuitable for AVS. Therefore, it is necessary to tailor a new method for AVS.

AVSBench (Zhou et al., 2022) is the first baseline method for AVS, but its convolutional structure limits its performance due to limited receptive fields (Gao et al., 2023). As a remedy, transformers have recently been introduced to AVS network designs (Gao et al., 2023; Liu et al., 2023c;b; Huang et al., 2023) by using the attention mechanism to model the audio-visual relationship. The attention mechanism can highlight the most relevant video frame regions for each audio input by aggregating the visual features according to the audio-guided attention weights. Despite some improvements, present AVS methods still suffer modality imbalance issues, as illustrated in Figure 1. Notably, modality imbalance generally exists in multi-modality fields (Han et al., 2022; Peng et al., 2022; Zhou et al., 2020).

In present AVS methods, audio features are insufficiently fused because the fusion is unidirectional and deficient, while the vision information is more sufficiently exploited (Zhou et al., 2022; Gao et al., 2023; Liu et al., 2023c;b). The inadequate audio feature either repeats itself to the same size as the visual feature for cross attention or serves as a query for channel attention in the fusion modules

Figure 1: Overview of the AVSBench baseline (Zhou et al., 2022), AVSegFormer (Gao et al., 2023) and our AVSAC. The fusion of audio features is insufficient in (a) and (b) because of their undirectional and deficient interaction, while the vision information is more sufficiently exploited. This will cause modality imbalance with visual features taking the dominance. (c) Our AVSAC features a paralleled decoder branch structure with multiple bidirectional bridges linked between each layer to strengthen audio cues through continual and in-depth audio-visual interaction, which relieves modality imbalance. It has two key designs: (1) Bidirectional Audio-Visual Decoder (BAVD), which has a two-tower decoder structure with each layer linked by bidirectional bridges, and (2) Audio Feature Reconstruction (AFR) module to recover lost audio representations from the output from our audio-guided vision decoder branch for better audio signal preservation.

of Figure 1 (a), (b) to produce what we define the audio-guided vision features (AGV). This implies a modal imbalance issue: the network will focus more on the visual information while neglecting the audio ones because the audio feature is not as information-rich as the visual feature, so unidirectional and deficient fusion of audio features may easily cause audio cues to fade away. This phenomenon resembles Liu et al. (2023a). This will restrict audio-visual representation learning as audio cues also matter to AVS. Thus, it's intuitive to derive vision-guided audio features (VGA) to balance with AGV by aggregating the audio features together according to the vision-guided attention weights. However, both AGV and VGA are single-modal features that only represent part of the multi-modal information. For instance, VGA is a set of audio features that describes pixels but does not preserve the inherent visual feature of each pixel itself. We argue that a holistic understanding of audio-visual representation can be obtained through the bidirectional interaction of the two modalities.

Motivated by this, we devise AVSAC, where we propose a bidirectional audio-visual decoder with a two-tower structure linked by multiple bidirectional bridges between each decoder layer. It integrates AGV and VGA into one structure, as shown in Figure 1 (c). The vision-guided decoder branch processes and outputs the VGA, while the audio-guided decoder branch processes and outputs AGV. The bidirectional bridges enable continuous and in-depth interaction between the two feature modes to boost the significance of audio representations.

In addition, since most AVS methods use the GT mask as the only supervision, no effective feedback can be given to prevent the network from learning a certain data bias Chen et al. (2023) that enables the network to "guess" the correct target even if the audio cue has been lost. We hence propose the Audio Feature Reconstruction (AFR) to protect network from harmful data bias and guarantee the existence of audio cues. AFR loss is then introduced to supervise network training.

Our main **contributions** can be summarized as three-fold: **(i)** We propose AVSAC, a two-tower decoder AVS structure with multiple bidirectional bridges linked between, to solve the modality imbalance issue in present AVS networks. **(ii)** We propose two modules: Bidirectional Audio-Visual Decoder (BAVD) and Audio Feature Reconstruction (AFR), to further enhance audio cues through continuous and in-depth audio-visual interaction while preventing the network from learning harmful data bias. **(iii)** Our method achieves new state-of-the-art performances on two sub-tasks of the AVS benchmark, with significant improvements (about 6% in mIoU and 4% in F-score) in the most challenging multiple sound source subset.

## 2 RELATED WORKS

### 2.1 AUDIO-VISUAL SEGMENTATION

Audio-visual segmentation (AVS) requires to predict pixel-wise masks of the sounding objects in a video sequence given audio cues. To tackle this issue, (Zhou et al., 2022; 2023) proposed the first

audio-visual segmentation benchmark with pixel-level annotations and devised AVSBench based on the temporal audio-visual fusion with audio features as query and visual features as keys and values. However, the relatively limited receptive field of convolutions restricts the AVS performance. As a remedy, AVSegFormer (Gao et al., 2023) was later proposed based on vision transformer and achieved good AVS performances. Following (Gao et al., 2023), a series of ViT-based AVS networks has also been proposed(Liu et al., 2023c;b; Huang et al., 2023). Recently (Mao et al., 2023) has introduced diffusion model to AVS. However, present methods do not pay enough attention to audio cues as to visual features, since the audio feature is not as information-rich as visual feature but the fusion of audio features is relatively insufficient and unidirectional, while the vision information is more sufficiently exploited by sending to every decoder layers. Therefore, modality imbalance occurs with more focus on the visual information, making audio cues easily fade away. To solve this, we propose to continuously strengthen audio cues to reach modality balance.

(Liu et al., 2023a) also noticed the modality imbalance problem but in a language-vision task, but its audio and visual feature extraction process only involves vision modality while we involves audio and vision modalities.

## 2.2 Vision Transformer

Drawing inspirations from Deformable DETR (Zhu et al., 2020) and DINO (Zhang et al., 2022), AVSegFormer (Gao et al., 2023) leveraged vision transformer to extract global dependencies. The remarkable performance of AVSegFormer (Gao et al., 2023) has inspired more researchers to apply transformer to AVS task (Liu et al., 2023c;b; Huang et al., 2023). However, the fusion of audio features is insufficient in these methods because it is unidirectional and unenough, while the vision information is more sufficiently exploited by sending it to every decoder layer for attention operations. This inevitably triggers modality imbalance–the network tends to focus more on the vision information instead of the little bits of audio cues, making the audio information easily fade away. To solve this, we devise AVSAC based on ViT to enhance the disadvantaged audio information so as to balance with visual features.

## 3 METHOD

### 3.1 OVERVIEW

Figure 2 illustrates the overall architecture of our AVSAC, which is an AVS network with a two-tower decoder structure linked with multiple bidirectional bridges. The network's inputs include an audio input $A \in \mathbb{R}^{T \times d_{model}}$ and consecutive video frames input $V$, where $T$ represents the number of frames and $d_{model}$ defaults to 128 in the VGGish. Following previous works (Zhou et al., 2022; Gao et al., 2023), we first extract two sets of input backbone features: visual feature $F_{vis}$ from $V$ using a CNN backbone ((He et al., 2016; Wang et al., 2022)), and audio feature $F_{audio}$ from $A$ using VGGish (Hershey et al., 2017). Then $F_{audio}$ and $F_{vis}$ are fed into corresponding decoders for multi-model interaction and output binary segmentation masks. Our AVSAC has two key components: (1) a Bidirectional Audio-Visual Decoder (BAVD) and (2) an Audio Feature Reconstruction (AFR) module. We detail each component of our framework in the following.

### 3.2 BIDIRECTIONAL AUDIO-VISUAL DECODER (BAVD)

As mentioned above, most present AVS methods use the naive attention mechanism to process multi-modal information but always use audio features as query and visual features as key and value through channel attention or repeating audio features to the same size as visual features. In this way, the audio features only attend the generation of attention weights that indicate the significance of each area in the visual feature but do not directly involve in the output, so that the output can be regarded as a reorganized single-modal vision feature, which we define as audio-guided vision features (AGV). Even worse is that the AGV is then sent to the successive transformer decoder. As a result, visual features take the dominance while audio features dramatically get lost in the decoder. Thus we argue that present AVS methods only do well in processing features from the value input, but lack the ability to fuse multi-modal features. We call this issue a modality imbalance challenge.

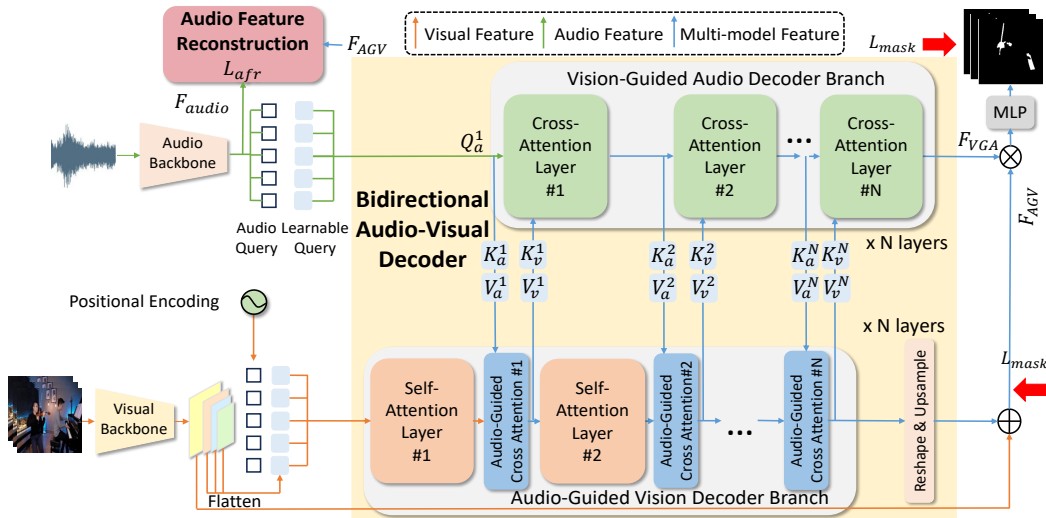

Figure 2: Overall architecture of AVSAC. We propose two key components in this framework: (1) Bidirectional Audio-Visual Decoder (BAVD), enabling the model to strengthen audio cues by consistently balancing AGV with VGA through continuous and in-depth bilateral modality interaction. (2) Audio Feature Reconstruction (AFR) module is proposed to evade harmful data bias and prevent audio representations from fading away.

To relieve modality imbalance, we hope to equally treat features from both modalities and interact with each other. Based on this idea, we propose a Bidirectional Audio-Visual Decoder (BAVD) that consists of two paralleled interactive decoders with bidirectional bridges, as shown in Figure 2. The two-tower decoder structure includes an audio-guided decoder branch and a vision-guided decoder branch, responsible for extracting AGV and VGA respectively. In this way, both AGV and VGA can be consistently extracted and mutually balance each other as the network goes deep, which significantly relieves modality imbalance. To obtain the final segmentation mask prediction, we multiply the AGV feature output $F_{AGV}$ obtained from the audio-guided vision decoder branch with the VGA feature output $F_{VGA}$ from the vision-guided audio decoder branch. Then we utilize an MLP to integrate different channels, followed by a residual connection to ensure that the fusion of audio representation does not lead to aggressive loss of visual representation. Finally, the segmentation mask $M$ can be obtained through a linear layer. The process be expressed as Equation 1.

$$M = Linear(F_{AGV} + MLP(F_{AGV} \cdot F_{VGA})) \tag{1}$$

where $MLP(.)$ and $Linear(.)$ denote the MLP process and the linear layer, respectively. In the following, we will introduce the two decoder branches in our BAVD in detail on how to obtain $F_{VGA}$ and $F_{AGV}$.

### 3.2.1 AUDIO-GUIDED VISION (AGV) DECODER BRANCH

We adopt the deformable attention block following (Zhu et al., 2020) as the basic structure of the self-attention layer in AGV decoder branch to generate visual queries for audio-guided cross attention in the next step, as shown in the right part of Figure 3 (a). We build multiple bidirectional bridges between each layer of the two decoders for bidirectional modality interaction. Specifically, the inputs of i-th layer of the VGA decoder branch are injected into the i-th layer of the AGV decoder branch as key ($K_a^i$) and value ($V_a^i$) for multi-modality interaction based on the cross-attention operation. The expression of audio-guided cross attention on the i-th layer is written as Equation 2.

$$F_{AGV}^i = CrossAtten(Q_v^i, K_a^i, V_a^i, F_v^i, w) = F_v^i + softmax(\frac{Q_v^i K_a^{i^T}}{\sqrt{d_K}})V_a^i w \tag{2}$$

where $F_{AGV}^i$ is the i-th audio-guided cross attention output, $K_a^i, V_a^i$ are the key and value sent through the bridge from the i-th layer of the VGA decoder branch and $Q_v^i$ is the query of the i-th

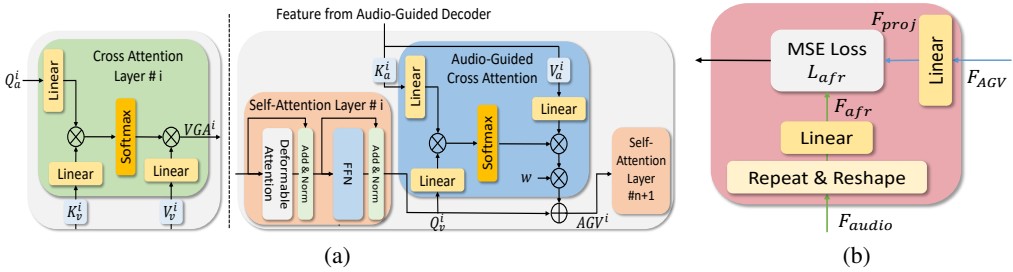

Figure 3: illustration of the Internal module structure of (a) Bidirectional Audio-Visual Vision Decoder (BAVD). (b) Audio Feature Reconstruction (AFR).

layer of the AGV decoder branch. $F_v^i$ is the i-th layer self attention output. $w$ is a learnable weight that enables the AGV decoder branch to determine how much information it needs and also makes the audio representation more adaptable to the audio-visual representation.

### 3.2.2 VISION-GUIDED AUDIO (VGA) DECODER BRANCH

The basic operation in our VGA decoder branch is the cross-attention operation, as shown in the left part of Figure 3 (a). We do not follow the same structure as AGV decoder branch as we delete the self-attention layer because we find that the audio representation is not as information-rich as the visual representation, and additional self-attention layers only bring increased parameters but dropped performance. The experimental result is recorded in our ablation study in Table 3. The reason is that excessive parameters may cause ambiguity in audio representations to harm the performance. Therefore we simplify our VGA decoder branch to only cross attention layers. The expression of the cross attention on the i-th layer is written as Equation 3.

$$F_{VGA}^i = CrossAtten(Q_a^i, K_v^i, V_v^i) = softmax(\frac{Q_a^i K_v^{i^T}}{\sqrt{d_k}})V_v^i, \quad Q_a^i = F_{VGA}^{i-1} \tag{3}$$

where $F_{VGA}^i$ is the i-th layer cross attention output. $Q_a^i$ is the query of the i-th layer of the VGA decoder branch, which is also the former layer cross-attention output. and $K_v^i, V_v^i$ are the key and value come from the i-th layer outputs of the AGV decoder branch.

### 3.3 AUDIO FEATURE RECONSTRUCTION (AFR)

Most of the present AVS networks are only supervised by the segmentation GT masks, which implies a hypothesis: as long as the prediction mask matches the GT mask, we think that the network can understand the audio cues. However, this is not always the case actually. For example, for some datasets, there is always one and only one target in the segmentation GT mask for each training sample, so the network may easily learn such data bias and always output one target. Thus, for some training samples, if the network happens to "guess" the right target region even if the audio information has been lost or distorted, these samples actually contribute nothing or even do harm to the network training.

To prevent the network from harmful data bias during representation learning and also curtail audio information loss, we propose the Audio Feature Reconstruction (AFR) module as shown in Figure 3 (b). AFR can reconstruct the audio feature from the final output of the audio-guided vision decoder to guarantee that the audio information is well preserved through the whole audio-visual feature interaction procedure. It takes audio feature $F_{audio}$ and the AGV output $F_{AGV}$ of the audio-guided vision decoder as inputs and projects the two features into the same feature space for comparison. The Audio Feature Reconstruction loss is derived by minimizing the distance between the reconstructed audio feature $F_{afr}$ and the projected multi-model feature $F_{proj}$ using the Mean Squared Error (MSE) loss.

### 3.4 Loss Function

The total loss function $\mathcal{L}$ consists of two parts: one is the mask loss for segmentation ($\mathcal{L}_{mask}$) that includes binary Focal loss $\mathcal{L}_{focal}$ (Lin et al., 2017) and Dice loss $\mathcal{L}_{dice}$ (Milletari et al., 2016), and the other is for audio feature reconstruction ($\mathcal{L}_{afr}$). We define $\mathcal{L}_{mask} = \lambda_{focal}\mathcal{L}_{focal} + \lambda_{dice}\mathcal{L}_{dice}$ and $L_{afr} = MSE(F_{afr}, F_{proj})$. We use the mask loss to supervise two places of the network–the final output and the audio-guided decoder output. The total loss function can be written as

$$\mathcal{L}(\hat{y}, y) = \sum_{i=1}^{2} \mathcal{L}_{mask}^i + \lambda_{afr}\mathcal{L}_{afr} = \sum_{i=1}^{2}(\lambda_{focal}\mathcal{L}_{focal}^i + \lambda_{dice}\mathcal{L}_{dice}^i) + \lambda_{afr}\mathcal{L}_{afr} \qquad (4)$$

where $\lambda_{focal}$, $\lambda_{dice}$ and $\lambda_{afr}$ are the weights to balance the loss function. The whole framework is trained by minimizing the total loss function between $\hat{y}$ and ground-truth $y$. Our proposed AFR does not participate in mask prediction and is computationally free during inference. It can serve as a plug-in module to any existing AVS methods.

## 4 Experiments

### 4.1 Datasets

**AVSBench-Object** (Zhou et al., 2022) is an audio-visual segmentation dataset with pixel-level annotations. Each video has a duration of 5 seconds and is uniformly segmented into five clips. Annotations for the sounding objects are provided by the binary mask of the final frame of each video clip. The dataset includes two subsets according to the number of audio sources in a video: a semi-supervised single sound source subset (S4), and a fully supervised multi-source subset (MS3). The S4 subset consists of 4932 videos, while the MS3 subset contains 424 videos. We follow the dataset split of (Zhou et al., 2022) for training, validation, and testing.

### 4.2 Evaluation Metrics

We employ the evaluation metrics including the Jaccard index (the mean Intersection-over-Union between the predicted masks and the ground truth) and F-score following (Zhou et al., 2022).

### 4.3 Implementation Details

We train our AVSAC model for the three sub-tasks using 2 NVIDIA A6000 GPUs. We freeze the parameters of visual and audio backbones. We choose the Deformable Transformer (Zhu et al., 2020) block as the structure of the self-attention layer in the vision-guided decoder. Consistent with previous works, we choose AdamW (Loshchilov & Hutter, 2017) as our optimizer, with the batch size set to 4 and the initial learning rate set to $2 \times 10^{-5}$. All video frames are resized to $224 \times 224$ resolution. For the S4 and MS3 subsets, each video contains 5 frames. Since the MS3 subset is quite small, we train it for 60 epochs, while the S4 is trained for 30 epochs. The encoder and decoder in our AVSAC both are comprised of 6 layers with an embedding dimension of 256. We set $\lambda_{focal}$, $\lambda_{dice}$ and $\lambda_{afr}$ to 1, 1 and 0.1 for the best performance.

### 4.4 Qualitative Comparison

We compare the qualitative results between AVSBench (Zhou et al., 2022), AVSegFormer (Gao et al., 2023) and our AVSAC in Figure 4. PVT-v2 is adopted as the visual encoder backbone of all three methods, among which our AVSAC enjoys the best visual segmentation performance with the finest shape-aware segmentation effect and significantly relieves some false alarms. The left side of Figure 4 visualizes the fineness of our method in delineating the shape of the sounding object. The right side of Figure 4 visualizes that our AVSAC can curtail some false alarms. When the pianist appears in the video frames but does not make any sound, AVSBench cannot find the correct-sounding objects and includes all of them. AVSegFormer performs better but still mistakes the person in the background as the sounding source in the third frame. However, our AVSAC can accurately segment the sounding piano keys without generating false alarm predictions. The reason

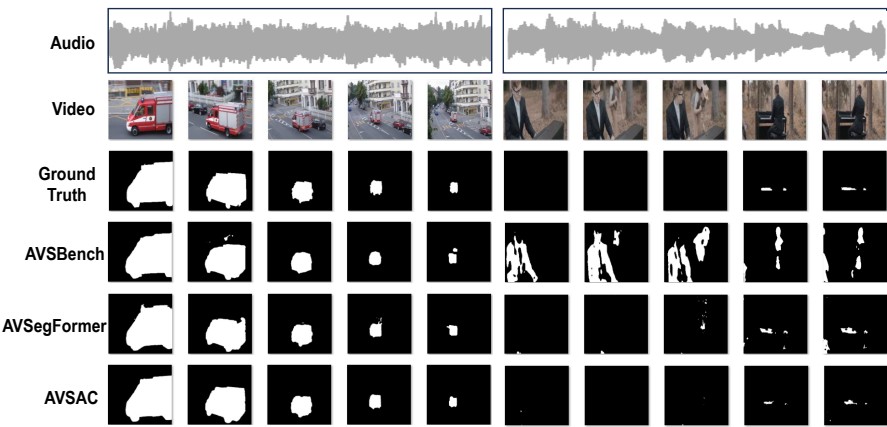

Figure 4: Qualitative examples of the AVSBench, AVSegFormer and our AVSAC framework. The left part is the output of the S4 setting, and on the right is the MS3 setting. The other two methods only produce segmentation maps that are not that precise, whereas our AVSAC can not only significantly evade some false alarms but also more accurately delineate the shapes of sounding objects.

for the flaws of the other two methods (also other AVS methods) is that the features extracted by these methods are dominated by visual representations while lacking enough audio guidance to help distinguish the correct-sounding object. Our method solves this modality imbalance problem by balancing visual representations with enough audio cues by building bidirectional bridges between our two-tower decoders so that both AGV and VGA can be consistently obtained.

We also visualize the attention maps of consecutive frames with and without audio enhancement in Figure 5 to further prove the effectiveness of our method. It is obvious to find that no-audio-enhancement results are coarser than audio-enhanced ones from the left part of Figure 5. The right part of Figure 5 again visualizes the false alarms caused by modality imbalance–the left man is mistaken as a sounding object even though he does not make any sound. Equipped with our bidirectional bridges between two decoders for continuous audio enhancement can significantly mitigate this issue. For more qualitative results, please refer to our appendix.

## 4.5 QUANTITATIVE COMPARISON

The quantitative results on the test set of S4 and MS3 are presented in Table 1. We can observe that our method has achieved state-of-the-art performances on all three AVS benchmark subsets in terms of F-score and mIoU. There is a consistent performance gain of our AVSAC model compared to other SOTA methods, regardless of whether ResNet-50 (He et al., 2016), Swin-Transformer (Liu et al., 2021) or PVTv2 (Wang et al., 2022) is used as the backbone. This suggests that the modality imbalance problem indeed exists and limits the performance of present AVS methods. By building multiple bidirectional bridges between two paralleled decoders, we balance audio and visual modality information flows during the whole training process, therefore solving the modality imbalance issue and significantly boosting the segmentation performances. Note that the performance improvement benefits from our two-tower decoder structure with bidirectional bridges itself, instead of coming from more trainable parameters. Actually, the parameter number of our baseline AVSegFormer (Gao et al., 2023) is 186M, while our model is slightly more lightweight with 181M parameters, but our model far outperforms present AVS methods.

## 5 ABLATION STUDIES

In this section, we conduct ablation experiments to verify the effectiveness of each key design in our proposed AVSAC. Specifically, we adopt PVTv2 (Wang et al., 2022) as the backbone and conduct extensive experiments on the S4 and MS3 sub-tasks.

**Ablation of each key module.** We ablate the impact of each of our key components in Table 2. We adopt AVSegFormer (Gao et al., 2023) as the baseline. From Table 2 we can observe that compared

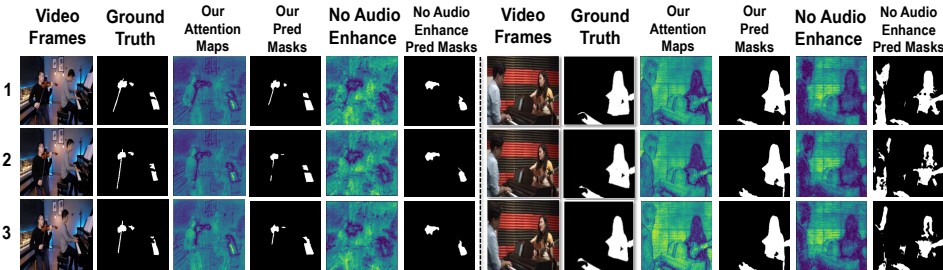

Figure 5: Visualization of attention maps of consecutive frames with and without multiple bidirectional bridges. The darker areas highlight the predicted-sounding objects in attention maps.

Table 1: Comparison with other SOTA methods on the AVS benchmark. All methods are evaluated on two AVS sub-tasks, including single sound source segmentation (S4) and multiple sound source segmentation (MS3). The evaluation metrics are F-score and mIoU. The higher the better. The figures in bold mark the highest ones in each column.

| Methods | Backbone | S4 | | MS3 | |
|---|---|---|---|---|---|
| | | F-score | mIoU | F-score | mIoU |
| LVS (Chen et al., 2021) | ResNet-50 (He et al., 2016) | 51.0 | 37.94 | 33.0 | 29.45 |
| MSSL (Qian et al., 2020) | ResNet-18 (He et al., 2016) | 66.3 | 44.89 | 36.3 | 26.13 |
| 3DC (Mahadevan et al., 2020) | ResNet-34 (He et al., 2016) | 75.9 | 57.10 | 50.3 | 36.92 |
| SST (Duke et al., 2021) | ResNet-101 (He et al., 2016) | 80.1 | 66.29 | 57.2 | 42.57 |
| AOT (Yang et al., 2021) | Swin-B (Liu et al., 2021) | 82.8 | 74.2 | 0.050 | 0.018 |
| iGAN (Mao et al., 2021) | Swin-T (Liu et al., 2021) | 77.8 | 61.59 | 54.4 | 42.89 |
| LGVT (Zhang et al., 2021) | Swin-T (Liu et al., 2021) | 87.3 | 74.94 | 59.3 | 40.71 |
| AVSBench-R50 (Zhou et al., 2022) | ResNet-50 (He et al., 2016) | 84.8 | 72.79 | 57.8 | 47.88 |
| AVSegFormer-R50 (Gao et al., 2023) | ResNet-50 (He et al., 2016) | 85.9 | 76.45 | 62.8 | 49.53 |
| AVSBG-R50 (Hao et al., 2023) | ResNet-50 (He et al., 2016) | 85.4 | 74.13 | 56.8 | 44.95 |
| AuTR-R50 (Liu et al., 2023c) | ResNet-50 (He et al., 2016) | 85.2 | 75.0 | 61.2 | 49.4 |
| DiffAVS-R50 (Mao et al., 2023) | ResNet-50 (He et al., 2016) | 86.9 | 75.80 | 62.1 | 49.77 |
| AQFormer-R50 (Huang et al., 2023) | ResNet-50 (He et al., 2016) | 86.4 | **77.0** | **66.9** | **55.7** |
| AVSAC-R50 (Ours) | ResNet-50 (He et al., 2016) | **86.95** | 76.90 | 65.11 | 51.13 |
| AVSBench-PVT (Zhou et al., 2022) | PVTv2 (Wang et al., 2022) | 87.9 | 78.74 | 64.5 | 54.00 |
| AVSegFormer-PVT (Gao et al., 2023) | PVTv2 (Wang et al., 2022) | 89.9 | 82.06 | 69.3 | 58.36 |
| AVSBG-PVT (Hao et al., 2023) | PVTv2 (Wang et al., 2022) | 90.4 | 81.71 | 66.8 | 55.10 |
| AuTR-PVT (Liu et al., 2023c) | PVTv2 (Wang et al., 2022) | 89.1 | 80.4 | 67.2 | 56.2 |
| DiffAVS-PVT (Mao et al., 2023) | PVTv2 (Wang et al., 2022) | 90.2 | 81.38 | 70.9 | 58.18 |
| AQFormer-PVT (Huang et al., 2023) | PVTv2 (Wang et al., 2022) | 89.4 | 81.6 | 72.1 | 61.1 |
| AVSAC-PVT (Ours) | PVTv2 (Wang et al., 2022) | **91.62** | **84.29** | **74.86** | **64.13** |

with our AFR, our BAVD brings more significant AVS performance gain, especially in the more challenging MS3 subset. The reason is that the bidirectional bridge design in BAVD continuously strengthens the audio feature along the decoder to guarantee that the audio feature will not fade away. In this way, the previously dominant visual features can be balanced by strengthened audio signals, which helps the network to more accurately delineate sounding object shapes according to the strengthened audio guidance. Our AFR is more like an auxiliary module to prevent the network from learning harmful data bias and recover some lost audio representations from visual signals, so equipped with it also brings performance gains.

**Ablation of BAVD.** The ablation studies of the internal structure of BAVD are presented in Table 3. In this work, we propose BAVD, which is a two-tower decoder structure with bidirectional bridges to enhance audio cues to reach a continuous audio-visual modality balance and interaction. To demonstrate the necessity of bidirectional bridges, we compare this design with two kinds of uni-

directional bridges–the audio-to-vision bridge (Model #1) and the vision-to-audio bridge (Model #2). The audio-to-vision bridge brings higher performance gain than the vision-to-audio bridge, which proves that strengthening audio cues as visual guidance is very effective in relieving modality imbalance. The network equipped with bidirectional bridges offers the best performance, suggesting that this design adequately balances audio and visual modalities. We also ablate the necessity of attention blocks in both decoder branches by deleting each AGCA in AGV decoder branch (Model #3) and adding a self-attention between each VGA decoder branch layers (Model #4). Results show that adding self-attention between each VGA layer slightly harms AVS fineness because the audio feature itself is not as information-rich as visual features, so processing it with excessive parameters may cause ambiguity, so we do not add a self-attention block to the audio decoder. Deleting AGCA in AGV decoder drops the performance, so we keep it.

**Ablation of the loss function.** We ablate the impact of the loss function in Table 4. The results indicate that (1) Combining Focal loss with Dice loss as the mask loss contributes better to segmentation fineness, because the Dice loss solves the foreground-background imbalance problem but ignores a further imbalance between easy and difficult samples, whereas Focal loss can focus on these difficult misclassified samples. (2) Equipping mask loss with our proposed AFR loss brings the best results because our AFR loss can prevent the network from learning some harmful data bias and also help reconstruct some lost audio cues to guarantee modality balance.

**Ablation of the number of decoder layers.** We also adjust the number of BAVD layers $N$ and show the results in Table 5. By increasing BAVD layer number from 2 to 6, we can observe consistent AVS performance gain. The best results on both subsets are obtained when $N$ is set to 6, which indicates that more BAVD layers bring better performances. The reason is that audio cues cannot get sufficiently enhanced if the decoder layer is not enough and, therefore cannot effectively mitigate the audio-visual modality imbalance problem, leading to relatively limited performances.

Table 2: Ablation study of each key design. Results show that BAVD and AFR complements each other and combining both performs better.

| No. | Settings | S4 | | MS3 | |
|---|---|---|---|---|---|
| | | F-score | mIoU | F-score | mIoU |
| #1 | Baseline | 89.9 | 82.06 | 69.3 | 58.36 |
| #2 | BAVD | 90.93 | 83.65 | 74.12 | 63.46 |
| #3 | BAVD+AFR | **91.62** | **84.29** | **74.86** | **64.13** |

Table 3: Ablation study of BAVD. It shows that building bidirectional bridges between two-tower decoders improves more.

| No. | Settings | S4 | | MS3 | |
|---|---|---|---|---|---|
| | | F-score | mIoU | F-score | mIoU |
| #1 | Audio → Vis | 90.64 | 83.28 | 72.86 | 61.98 |
| #2 | Vis → Audio | 90.12 | 82.46 | 70.03 | 58.89 |
| #3 | w/o AGCA | 90.75 | 83.41 | 73.22 | 62.58 |
| #4 | +SA in VGAD | 91.01 | 83.86 | 73.66 | 62.99 |
| #5 | w/ BAVD | **91.62** | **84.29** | **74.86** | **64.13** |

Table 4: Ablation study of loss function. It shows that combining mask loss (Focal+Dice) with audio reconstruction loss works better.

| Settings | S4 | | MS3 | |
|---|---|---|---|---|
| | F-score | mIoU | F-score | mIoU |
| $\mathcal{L}_{mask}$ (Dice) | 90.56 | 83.29 | 73.66 | 62.99 |
| $\mathcal{L}_{mask}$ (Dice+Focal) | 90.93 | 83.65 | 74.12 | 63.46 |
| Dice+Focal+$\mathcal{L}_{afr}$ | **91.62** | **84.29** | **74.86** | **64.13** |

Table 5: Ablation study of decoder layer number. We find that 6 decoder layers work better than other configurations.

| Layers | S4 | | MS3 | |
|---|---|---|---|---|
| | F-score | mIoU | F-score | mIoU |
| 2 | 88.76 | 79.83 | 69.77 | 58.62 |
| 4 | 90.50 | 82.84 | 72.17 | 61.25 |
| 6 | **91.62** | **84.29** | **74.86** | **64.13** |

## 6 CONCLUSION

In this paper, we notice that present AVS methods all suffer a modality imbalance challenge–visual features take the dominance while audio signals easily fade away and are unable to provide enough guidance, causing relatively limited AVS performance. To solve this, we try to bootstrap AVS task by strengthening the audio cues to make a balance between audio and visual modalities and propose the AVSAC network. The superior performances on all three AVS datasets demonstrate the effectiveness of our method in solving modality imbalance in AVS.

**Limitations.** Similar to other methods that adopt Transformers, our proposed method does not exhibit significant advantages in model size and inference efficiency compared to recent AVS methods.

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

# A APPENDIX

## A.1 MORE QUALITATIVE RESULTS

Some more qualitative results between AVSBench (Zhou et al., 2022), AVSegFormer (Gao et al., 2023) and our AVSAC are visualized in Figure 6. We also adopt PVT-v2 as the visual encoder backbone of all three methods. Our AVSAC still enjoys the best visual segmentation performance with the finest shape-aware segmentation results.

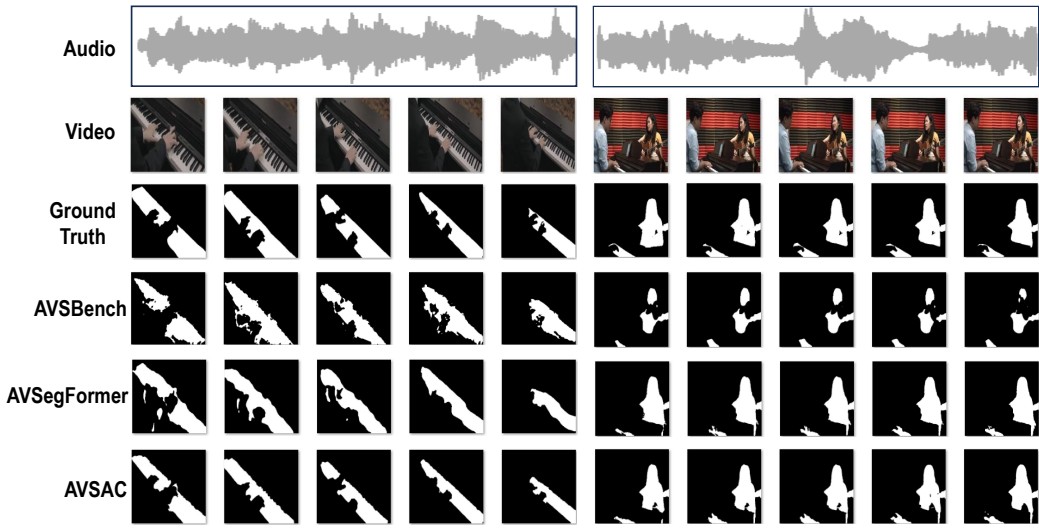

Figure 6: More Qualitative examples of the AVSBench, AVSegFormer, and our AVSAC. The left part is the output of the S4 setting, and on the right is the MS3 setting. The other two methods only produce segmentation maps that are not that precise, whereas our AVSAC can not only significantly evade some false alarms but also more accurately delineate the shapes of sounding objects.

## A.2 QUANTITATIVE RESULTS ON AVSS BENCHMARK

**AVSS** (Zhou et al., 2023) is an extension of the AVSBench-Object dataset with 12,356 videos across 70 categories. It is designed for audio-visual semantic segmentation (AVSS). Different from the AVSBench-Object dataset which only has binary mask annotations, AVSS offers semantic-level annotations for sounding objects. The videos in AVSS are longer, each lasting 10 seconds, and 10 frames are extracted from each video for prediction.

We add the quantitative results on the test set of AVSS based on the comparison results from (Zhou et al., 2023), as presented in Table 6. We can observe that our method still achieves the best performances on the larger AVS dataset in terms of F-score and mIoU.

## A.3 DETAILED SETTINGS

We list the detailed settings of our model in Table 7.

## A.4 FAILURE CASES

We visualize a failure case of present methods including our method in Figure 7 (a). When the real sounding object moves outside of the field of view, present AVS methods tend to falsely segment other object in view as sounding object.

## A.5 ABLATION OF TRAINING MASK LOSS W/ AND W/O SELF-ATTENTION

We display the training and evaluation mask losses between the final mask output and GT of AVSAC w/ and w/o self-attention in the VGA decoder branch in Figure 7 (b). Adding self-attention makes

Table 6: Comparison with other SOTAs on the AVSS benchmark. The evaluation metrics are F-score and mIoU. The higher the better. The figures in bold mark the highest ones in each column.

| Methods | Backbone | AVSS | |
|---|---|---|---|
| | | F-score | mIoU |
| 3DC (Mahadevan et al., 2020) | ResNet-34 | 21.6 | 17.27 |
| AOT (Yang et al., 2021) | Swin-B | 31.0 | 25.40 |
| AVSBench-R50 (Zhou et al., 2022) | ResNet-50 | 25.2 | 20.18 |
| AVSegFormer-R50 (Gao et al., 2023) | ResNet-50 | 29.3 | 24.93 |
| AVSAC-R50 (Ours) | ResNet-50 | **29.60** | **25.32** |
| AVSBench-PVT (Zhou et al., 2022) | PVTv2 | 35.2 | 29.77 |
| AVSegFormer-PVT (Gao et al., 2023) | PVTv2 | 42.0 | 36.66 |
| AVSAC-PVT (Ours) | PVTv2 | **42.25** | **36.90** |

Table 7: Detailed settings. This table provides a detailed overview of the specific settings used for each sub-task.

| Settings | S4 | MS3 | AVSS |
|---|---|---|---|
| input resolution | $224 \times 224$ | $224 \times 224$ | $224 \times 224$ |
| feames T | 5 | 5 | 10 |
| embedding dimension D | 256 | 256 | 256 |
| AGV decoder feature size | $5376 \times 256$ | $5376 \times 256$ | $1029 \times 256$ |
| VGA decoder feature size | $300 \times 256$ | $300 \times 256$ | $300 \times 256$ |
| decoder layers N | 6 | 6 | 6 |
| batch size | 4 | 4 | 4 |
| optimizer | AdamW | AdamW | AdamW |
| learning rate | $2 \times 10^{-5}$ | $2 \times 10^{-5}$ | $2 \times 10^{-5}$ |
| drop path rate | 0.1 | 0.1 | 0.1 |
| epoch | 30 | 60 | 30 |

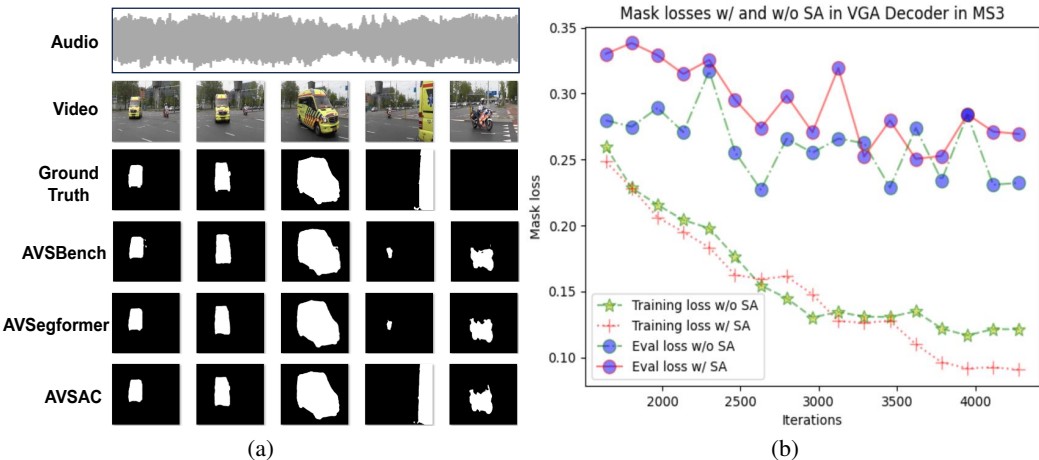

|     |     |
| :-: | :-: |
| (a) | (b) |

Figure 7: (a) Failure case display. (b) Training mask losses between the final mask output and GT of AVSAC w/ and w/o self-attention in the VGA decoder branch.

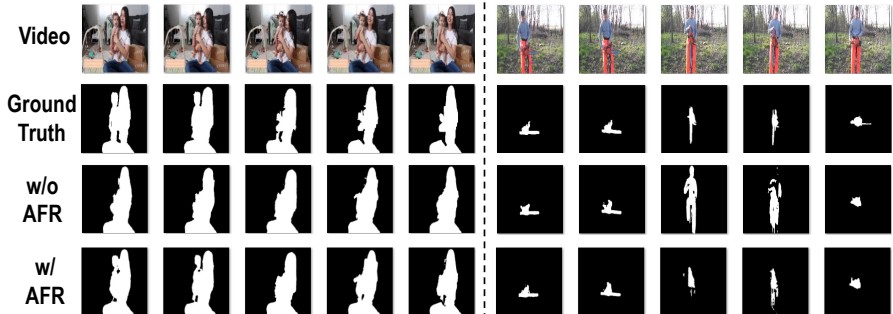

Figure 8: Visual ablation of AFR module.

the mask loss drops lower while the evaluation loss is higher than the no self-attention case, which means over-fitting. Therefore this explains why adding self-attention to VGA decoder is not adopted.

## A.6 VISUAL ABLATION OF AFR MODULE

We visualize visual result samples with and without AFR module in Figure 8. Take the left part result as an example, when we remove AFR, we find that the model cannot segment the laughing baby on the first two frames since part of the audio cues may get lost so that the audio guidance is not enough. Adding AFR module improves this because the sounding baby gets successfully recognized and segmented again.

## A.7 VISUAL RESULT WITHOUT AUDIO CUES

We visualize a visual result without audio cues in Figure 9. We use our pretrained AVSAC model but set the audio input as zero and obtain the no audio results in the third row. Obviously, without audio cues dramatically harms the AVS performance.

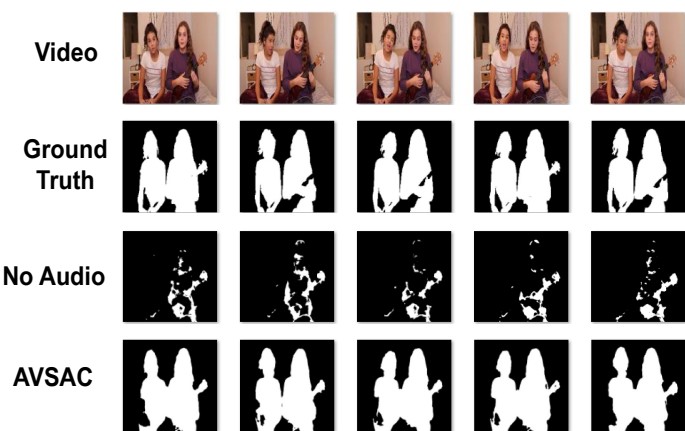

Figure 9: Visual result without audio cues.

