# OpenReview forum: "Bootstrapping Audio-Visual Segmentation by Strengthening Audio Cues"
_ICLR.cc/2024/Conference — Submitted to ICLR 2024_

### Official Review · Reviewer_xbK3 · 2023-10-26

**Soundness:** 2 fair
**Presentation:** 3 good
**Contribution:** 2 fair
**Rating:** 3
**Confidence:** 4

**Summary:**

The paper mainly studied the salient audio-visual segmentation problem (AVS) which tries to segment the sounding objects given the audio query. The paper argues that the current AVS methods generally suffer a modality imbalance issue and output features are always dominated by the visual representation hence hindering the model performance. To address such issues, the paper proposed a bidirectional audio-visual decoder (AVSAC) that builds mutual cross-attention layers between audio and visual streams. In addition, the audio feature reconstruction (AFR) evades harmful data bias and aims to preserve useful audio information. Extensive experiments show that the proposed methods achieve better results than the previous methods.

**Strengths:**

The paper is generally well-written and easy to follow.
The author provides extensive experiments to demonstrate the effectiveness of the model including both qualitative results and quantitative results.

**Weaknesses:**

My main concern regarding the paper's technical contribution is that the proposed bidirectional framework is not entirely novel and has been previously explored in multiple instances (e.g., [a, b, c]). These prior methods share a similar concept with the proposed approach, and the use of reconstruction to preserve the semantic meaning of the audio can also be viewed as a preliminary version of [d].

In the introduction, the author pointed out the issue of dataset bias in the current AVS dataset, which can potentially allow the network to make accurate predictions even when audio information is absent. This perspective aligns with the findings presented in [e], where a similar context of dataset bias was discussed. It would be great for the author to properly cite this prior work and discuss it if necessary.

The author did not include the SOTA method AQFormer [f] in the experiment section, which shows better results compared to the proposed method in Table 1. For example, AQFormer (ResNet50) achieved a 55.7 mIoU score under the MS3 split, while the proposed method achieved only 51.13. While it's understandable that the proposed method may not have surpassed the previous approach, but some discussion on this discrepancy would be appreciated. Similarly, when comparing the previous method on AVSS (Table 6). I suppose the AVSegformer's performance should be also listed in that table.



**Reference**

[a] Tao, R., Pan, Z., Das, R.K., Qian, X., Shou, M.Z. and Li, H., 2021, October. Is someone speaking? exploring long-term temporal features for audio-visual active speaker detection. In Proceedings of the 29th ACM International Conference on Multimedia (pp. 3927-3935).

[b] Duan, B., Tang, H., Wang, W., Zong, Z., Yang, G. and Yan, Y., 2021. Audio-visual event localization via recursive fusion by joint co-attention. In Proceedings of the IEEE/CVF Winter Conference on Applications of Computer Vision (pp. 4013-4022).

[c] Liu, S., Quan, W., Liu, Y. and Yan, D.M., 2022, May. Bi-directional modality fusion network for audio-visual event localization. In ICASSP 2022-2022 IEEE International Conference on Acoustics, Speech and Signal Processing (ICASSP) (pp. 4868-4872). IEEE.

[d] Georgescu, M.I., Fonseca, E., Ionescu, R.T., Lucic, M., Schmid, C. and Arnab, A., 2023. Audiovisual masked autoencoders. In Proceedings of the IEEE/CVF International Conference on Computer Vision (pp. 16144-16154).

[e] Chen, Y., Liu, Y., Wang, H., Liu, F., Wang, C. and Carneiro, G., 2023. A Closer Look at Audio-Visual Semantic Segmentation. arXiv e-prints, pp.arXiv-2304.

[f] Huang, S., Li, H., Wang, Y., Zhu, H., Dai, J., Han, J., Rong, W. and Liu, S., 2023. Discovering Sounding Objects by Audio Queries for Audio Visual Segmentation. arXiv preprint arXiv:2309.09501.

**Questions:**

Please refer to the comments in the weakness section.

---

> ### Author Response · Authors · 2023-11-17
> **Responses to Reviewer xbK3**
>
> Thank you for your valuable comments. Below are our responses.
>
> **Q1: The proposed bidirectional framework is not entirely novel and has been previously explored in [a,b,c]，and the use of reconstruction to preserve the semantic meaning of the audio can also be viewed as a preliminary version of [d].**
>
> **A1:** Our bidirectional framework is quite different from [a,b,c]. The novelty of our bidirectional framework (BAVD) lies in 3 aspects: bi-direction, continuity and motivation emphasis. The modality interaction is **bidirectional** and **continuous**, and the emphasis of our motivation is on **enhancing audio cues to evade modality imbalance**. [a,b,c] all focus on building better correlation between audio-visual features instead of on audio cues. Moreover, [a] interacts modalities only once so it lacks continuity; [b] continuously interacts modalities but it is unidirectional, which is not as sufficient as ours; [c] use different cross attention mechanism as ours and also its interaction is only once and unidirectional.
>
> Also, our AFR module is not a preliminary version of [d], since the two differs much in motivation, structure and task. The motivation of [d] is to use audiovisual information already present in video to improve self-supervised representation learning. To achieve this, [d] design an audiovisual masked autoencoder that can use audio and mask video tokens to reconstruct video tokens. However, the motivation of our AFR is to further curtail audio information loss. To achieve this, we design a linear module to use visual feature to reconstruct lost audio feature and propose a loss function to constrain this process. Also, our AFR’s task is audio-visual segmentation while the task of [d] is to improve self-supervised representation learning.
>
> **Q2: The dataset bias in this paper aligns with the finding in [e]. Properly cite this prior work and discuss it if necessary.**
>
> **A2:** We have cited this work and highlighted the place in yellow in our updated version. Notably, [e] and us adopt different ways to mitigate data bias. [e] discovered that AVS datasets are biased and proposed a relatively unbiased AVS benchmark to mitigate this issue from dataset level. While we think the bias may be caused by lack of enough audio guidance and seek to reconstruct lost audio cues with AFR module to mitigate the issue. Both of our solutions work well.
>
> **Q3: Include the SOTA method AQFormer in the experiment section and discuss the discrepancy. Also, list the AVSegformer's performance listed in table 6 when comparing on AVSS.**
>
> **A3:** Thank you for your question. We have added it in our updated pdf file and highlighted it in green color in Table 1. The reason why we did not compare with AQFormer is that this paper belongs to concurrent work, since it was newly released on arxiv on Sep.18.2023 and has not been officially published before. Here we show our comparison with AQFormer in Table. i., where our AVSAC generally surpasses AQFormer more significantly than AQFormer surpasses ours, despite the inferiority on MS3 with ResNet-50 as visual backbone.
>
> We provide AVSegformer's performance on AVSS on Table.ii where our AVSAC is superior. We have updated our pdf file and added this to Table. 6 in the appendix.
>
>
> Table. i: Comparison between AVSAC and AQFormer on two S4 and MS3.
> | Methods | Backbone | F-score (S4)| mIoU (S4)| F-score (MS3)| mIoU (MS3)|
> | --- | --- | --- | --- |--- | --- |
> | AQFormer-R50 | ResNet-50 | 86.40 | 77.00  | 66.90 | 55.70  |
> | ACSAC-R50 | ResNet-50 | 86.95|76.90|65.11|51.13|
> | AQFormer-PVT|PVT-v2|89.40|81.60|72.10|61.10|
> | ACSAC-PVT|PVT-v2|91.62|84.29 |74.86 |64.13 |
>
> Table. ii: Comparison between AVSegFormer and our AVSAC on AVSS.
> | Methods | Backbone | F-score| mIoU|
> | --- | --- | --- | --- |
> | AVSegFormer-R50 | ResNet-50 | 29.30|24.93|
> | AVSAC-R50 | ResNet-50 | 29.60|25.32|
> | AVSegFormer-PVT|PVT-v2|42.00|36.66 |
> | AVSAC-PVT | PVT-v2 | 42.25|36.90|

---

> ### Comment · Reviewer_xbK3 · 2023-11-17
> **Acknowledging the author's responses**
>
> Dear Authors,
>
> Thank you so much for the detailed explanation of my comments. The author has effectively addressed my concerns regarding Q2 and Q3. I agree with the author's decision for Q3, considering that it is not necessary to include AQFormer in the current version, given that it was a newly released paper.
>
> I appreciate the author highlighting the differences between the proposed work and the listed literature against Q1. However, I still believe that the proposed bidirectional architecture, linking between two modalities, offers sufficient novelty despite the high similarity mentioned previously. In addition to the literature mentioned earlier, there is also an audio-visual paper [g,h] that adopted the same bidirectional modulation, for example:
> \hat{f}_{t}^{a} = f^{a} + self-attention(f^{a},f^{a},f^{a}) + cross-attention(f^{a}, f^{v}, f^{v}),
> \hat{f}_{t}^{v} = f^{v} + self-attention(f^{v},f^{v},f^{v}) + cross-attention(f^{v}, f^{a}, f^{a}),
> and the claim about the cross-modality continuous interaction is the same as [I, h].
> In the referential expression segmentation paper [I], they proposed Language Feature Reconstruction (LFR), which shares a similar concept with the AFR module. The LFR module aims to prevent the loss or distortion of language information in the extracted features which is very similar to AFR. Additionally, the issue of modal imbalance has also been addressed in [h].
>
> In short, the proposed method and its motivation have significant overlap with the previous method [h] in terms of:
> 1) bidirectional/mutual attention and continuous, in-depth interactions
> 2) reconstruction of weak modality features to prevent information loss.
> 3) modal imbalance issue.
>
> Thus my original concern about novelty and originality of the proposed method remains unaddressed.
>
>
> [g] Lin, Y.B., Tseng, H.Y., Lee, H.Y., Lin, Y.Y. and Yang, M.H., 2021. Exploring cross-video and cross-modality signals for weakly-supervised audio-visual video parsing. Advances in Neural Information Processing Systems, 34, pp.11449-11461.
>
> [h] Liu, C., Ding, H., Zhang, Y. and Jiang, X., 2023. Multi-modal mutual attention and iterative interaction for referring image segmentation. IEEE Transactions on Image Processing.
>
> [I] Zhou, J., Wang, J., Zhang, J., Sun, W., Zhang, J., Birchfield, S., Guo, D., Kong, L., Wang, M. and Zhong, Y., 2022, October. Audio–visual segmentation. In European Conference on Computer Vision (pp. 386-403). Cham: Springer Nature Switzerland.

---

> ### Author Response · Authors · 2023-11-18
> **Responses to the reviewer's concerns**
>
> Dear reviewer:
>
> Thank you for your reply. We clarify our differences with your mentioned works in task, motivation, structure etc. below. Hope our responses may address your concerns.
>
> **Q1: Audio-visual paper [g] adopted the same bidirectional modulation, and the claim about the cross-modality continuous interaction is the same as [I, h].**
>
> **A1:** Our bidirectional module is very different from [g] since the modules are designed for different tasks, which also lead to different interaction modes.
>
> **Task:** The task of [g] is audio-visual video parsing, which is a categorization task that aims to temporally parse a video into audio or visual event categories. While our task is AVS, which needs audio guidance to help segmenting sounding targets in a video.
>
> **Interaction Mode:** The interaction of [g] is at a **inter-video** level, **not continuous**,  and **treats audio and visual signals equally**. [g] first aligns audio and video features and then does cross attention for interaction only once for each video. [g] also equally treats audio and video signals since its aim is modality interaction. However, the interaction mode of our bidirectional module is at **intra-video** level, **continuous** and **lays emphasis on audio cues**. We also **do not adopt self-attention in our audio branch as [g] does**. Our interactions are done in each video frames. We think audio cues should be emphasized and enhanced because it easily gets lost during training, otherwise other silent objects may be falsely segmented. We use enhanced visual features to extract audio signals useful to AVS from audio cues and then continuously feed these useful audio signals back to visual decoder, so we do not enhance audio cues again with the self-attention. Our experiment also proves that adding self-attention to our audio branch is harmful (Figure 7 (b) in our updated appendix).
>
> **Our cross-modality continuous interaction is also different from [I, h].** The interaction of [I] is unidirectional and cannot prevent audio cue loss, but ours is bidirectional and can prevent this. The cross-modality continuous interaction part of [h] is its multi-modal mutual decoder, but its structure is very different from our decoder BAVD since (1) [h]’s interaction is based on one branch while ours has two branches (2) [h]’s interaction needs to intake multi-stage vision features from encoder, while ours just needs to input the single visual feature output from visual encoder only once. The reason is that our parallel design is more efficient and makes multiple inputs not necessary.
>
> **Q2: The proposed method and its motivation have significant overlap with [h] in terms of:
> 1.bidirectional/mutual attention and continuous, in-depth interactions
> 2.reconstruction of weak modality features to prevent information loss.
> 3.modal imbalance issue.**
>
> **A2:** [h] solves the referring image segmentation (text-vision task), while we solve the AVS task. Text signals contain highly condensed semantic information and have no noise, while audio signals are not as information-rich as texts and may have noise. Since the modality differs, our designs also differs in motivation and structure:
>
> 1.The **motivation** of our bidirectional interaction design is to curtail audio-vision modality imbalance, while for [h] is to better fuse text-visual information. We achieve this by continuously feeding the useful audio cues to AVS extracted by visual features back to visual branch, while [h] turns to better interaction designs. For **structure**, [h] and our method **adopt different ways to conduct attentions** and **the attentions are conducted in different decoder structures**. Our attentions are conducted on two parallel decoders, while [h] implements cross attentions in one decoder. Also, the decoder of [h] takes multi-stage vision features as inputs to serve as k, v in cross attentions, while ours only intakes one visual input for self-attention and serve as q in our audio-guided cross attention. The reason is that our parallel design is more efficient and makes multiple inputs not necessary.
>
> 2.Our AFR module also differs from the reconstruction module of [h] in **motivation**, **structure** and **location**. The **motivation** of [h]'s module is to reconstruct text feature from visual feature, while our AFR is to reconstruct audio cues from visual features. For **structure**, our AFR module is simpler in structure without the position embedding and average pooling layers that [h]'s module has. For **location**, our AFR is located at the end of decoder to further guarantee our final visual output won’t lose audio cues, while [h]’s is placed between decoders.
>
> 3.Modality imbalance generally exists in many multi-modality tasks, and [h] notices the same issue as us despite modality difference. The differences of our solutions have been explained in the two points above.
>
> If you have any other concern, please reply at any time. Wish you all the best!

---

> > ### Author Response · Authors · 2023-11-21
> >
> > Dear Reviewer xbK3:
> >
> > Thank you for your valuable comments on our paper. We have explained the differences between our work with your mentioned papers. Could you let us know your thoughts on our newest responses? Hopefully, our responses can solve your concerns. If you have additional questions, we would be happy to answer.
> >
> > Sincerely, Authors

---

> ### Comment · Reviewer_xbK3 · 2023-11-21
> **Thanks for your response**
>
> Unfortunately, I remained unconvinced after the rebuttal. The M^3Dec [h] states clearly that they have noticed the modal imbalance issue in the attention-based network and suspect it is due to the characteristics of the Transformer’s decoder architecture. They further mention, "This implies a modal imbalance issue: the network will tend to focus more on the vision information, and the language information may be faded away during the information propagation along the network". This differs from the author’s response in Q2(1), where the design is simply described as better fusing text-visual information. Notably, the sentence in [h] exhibits significant similarity with the author’s claim on page 2 that "This implies a modal imbalance issue: the network will focus more on the visual information while neglecting the audio ones because the audio feature is not as information-rich as the visual feature, so unidirectional and deficient fusion of audio features may easily cause audio cues to fade away". However, the reference was not provided in the paper. For the feature reconstruction module, [h] tries to reconstruct the language feature from the last decoder layer’s output (multi-modal feature), which is similar to the AFR.
> In summary, the notable overlap between the proposed method and [h] raises doubts about the originality and distinctive contribution of the proposed model.

---

> > ### Author Response · Authors · 2023-11-22
> >
> > **Q1: The M^3Dec [h] states clearly that they have noticed the modal imbalance issue.**
> >
> > **A1:** The modality imbalance issue generally exists in multi-modality fields and has been studied by many works [a,b,c], also including [h]. When the modalities are different, the solutions also differ. Most methods adopt information interaction or enhancement to mitigate the imbalance, but the concrete solution differs as the modality feature differs. We have added these citations (highlighted in purple color) to our updated paper and claim the general existence of modality imbalance issue.
> >
> > **Q2: [h]’s design is simply described as better fusing text-visual information by the authors. Also, the sentence in [h] exhibits significant similarity with the author’s claim but the reference was not provided in the paper.**
> >
> > **A2:** [h] and us share the same target to mitigate modality imbalance, but our solution is different considering that our modalities are different. [h] adopts cross attentions to interact audio cues with multi-stage vision features for many times, **but its audio and visual feature extraction process only involves single modality**. For example, the multi-stage vision features are extracted from a ViT encoder without any audio cue participation. In contrast, **we propose to embed visual and audio signals to feature extraction process**. For example, at i-th layer in our BAVD, the visual feature first goes through a self-attention to enhance itself before interacting with audio cues in cross attention. The cross attention output is then fed back to the audio decoder branch for cross attention. After such a round of bidirectional interaction, the enhanced audio and visual features come to (i+1)-th layer and repeat the former operations. In this way, the interaction can be more efficient, and the decoder features can focus more on our interested sounding object areas. We have cited [h] in the related work section and added some discussions on our differences with [h] in our updated paper (highlight in pink color).
> >
> > **Q3: [h] tries to reconstruct the language feature from the last decoder layer’s output (multi-modal feature), which is similar to the AFR.**
> >
> > **A3:** [h] and our method have differences in reconstruction modules. (1) we reconstruct audio cues from AGV, which is of higher resolution than [h]’s last decoder layer’s output. (2) we repeat and reshape audio cues to align with visual features in every spatial location.
> >
> > Unfortunately, since [h] has no public code, we cannot directly migrate [h] to our task for comparison. If we have the source code, we would be happy to include the results.
> >
> > In summary, [h] and our method have many differences in motivation and structure, especially in structure. Hopefully our detailed analysis can remove your concern on our contribution and novelty.
> >
> >
> > [a] Han, Z., Zhang, C., Fu, H., & Zhou, J. T. “Trusted multi-view classification with dynamic evidential fusion.” TPAMI 2022.
> > [b] Peng, X., Wei, Y., Deng, A., Wang, D., & Hu, D. “Balanced multimodal learning via on-the-fly gradient modulation.” CVPR 2022.
> > [c] Zhou, K., Chen, L., & Cao, X. “Improving multispectral pedestrian detection by addressing modality imbalance problems.” ECCV 2020.

---

> ### Author Response · Authors · 2023-11-23
>
> Dear Reviewer xbK3,
>
> Sorry for bothering you again. Since **there is less than a day left until the end of the discussion**, we would like to further discuss with you to address your concerns about our overlaps with [h] in solving the similar modality imbalance issue, sentence similarity with [h] without reference and AFR’s similarity with [h]’s language reconstruction module in reconstruction from the last decoder layer’s output.
>
> As for the modal imbalance issue, we have already explained in our previous response modality imbalance issue generally exists in multi-modality fields and has been studied by many other works including [h]. Considering the modalities of our task are different from [h]’s, when the modalities change, the solutions also differ.
>
> Regarding the issue you raised about sentence similarity, we have updated our describtion in the paper. Furthermore, we have cited [h] in the related work section and added more discussions on our differences with [h] in our updated paper. Specifically, the audio and visual feature extraction process of [h] only involves single modality because the multi-stage vision features are extracted from a ViT encoder without any audio cue participation. While we propose to embed visual and audio signals to feature extraction process, so that the interaction can be more efficient, and the decoder features can focus more on our interested sounding object areas.
>
> For the AFR’s similarity with [h]’s language reconstruction module, we repeat and reshape audio cues to align with visual features (AGV) of higher resolution than [h]’s in every spatial location.
>
> In summary, we believe that our method differs significantly from h in both design and implementation aspects.
>
> Sincerely,  Authors

---

### Official Review · Reviewer_MwLf · 2023-10-28

**Soundness:** 3 good
**Presentation:** 2 fair
**Contribution:** 3 good
**Rating:** 6
**Confidence:** 3

**Summary:**

This paper presents a new method for audio-visual segmentation, via a Bidirectional Audio-Visual Decoder (BAVD) and Audio Feature Reconstruction (AFR). It addresses the problem of modality imbalance in audio-visual segmentation, claiming itself achieving new state-of-the-art results, including substantial improvements in challenging scenarios.

**Strengths:**

1. The paper's originality is properly presented and mentioned.
2. The paper has good level of clarity. The connection of its methods and performance is clearly presented. The backbone acquired is state-of-the-art level, although some of the baseline models are a bit old.

**Weaknesses:**

1. The use of English is a bit problematic. Please go through language checking (via tools like ChatGPT for example, if accessible) to fix some issues.
2. Since the author mention the data imbalance, the beginning of this paper somehow gives the impression to the reader that this paper is targeting this problem. However, after reading the paper, it is still a bit hard for the reviewer to find the problem being directly targeted. The method does lead to good level of improvement, but the imbalance problem gradually become secondary. Strengthening audio cues does give better performance, but that does not mean it tackles the imbalanced problems from my perspective. Also, whether a more balanced flow between video and audio will lead to better performance shall be discussed at the beginning, as part of motivation for the study.

**Questions:**

1. In the background, the imbalance issue between audio and video is mentioned, which is good. But I do not understand the aftermath of such imbalance - why the model's being focused more on visual information is a bad thing? This needs to be motivated a bit more.
2. Do you see any possibility that the architecture can be optimized to improve the efficiency? For example, some of the cross-modal connections might be redundant? Of course, here "nope" is also a proper answer.
3. At the beginning of Section 3.2 - what is the dot product in MLP() means in Equation 1?

**Details Of Ethics Concerns:**

This paper does not have ethical concern from the reviewer's perspective.

---

> ### Author Response · Authors · 2023-11-17
> **Responses to Reviewer MwLf**
>
> Thank you for your valuable comments. Below are our responses.
>
> **Q1: I do not understand the aftermath of such imbalance - why the model's being focused more on visual information is a bad thing? This needs to be motivated a bit more.**
>
> **A1:** Focusing more on visual information is ok but should be in moderation. Present AVS methods focus so much on visual information that neglects that some effective audio features will get lost during training and therefore cannot provide enough audio guidance for correct audio-visual segmentation of sounding objects. This can be visually observed from the right part example of Figure 4 in our paper, where irrelevant objects get mistaken as sounding objects because audio cues do not receive sufficient focus. After all, the AVS task is acoustic in nature as reviewer 7Z8k agrees.
>
> **Q2: Do you see any possibility that the architecture can be optimized to improve the efficiency?**
>
> **A2:** Yes, thank you for your valuable suggestion. More efficient and lightweight attention module, for example the squeeze-enhanced axial attention of SeaFormer [1], can be adopted to replace our used deformable attention, self-attention and cross attention modules to increase efficiency. Increasing model efficiency will be our future direction.
>
> **Q3：At the beginning of Section 3.2 - what is the dot product in MLP() means in Equation 1?**
>
> **A3:** It is the inner product of two metrics, multiplying $F_{VGA}$ of size 5 $\times$ 300 $\times$ 256 with $F_{AGV}$ of size 5 $\times$ 256 $\times$ 128 $\times$ 128 to get a new matrix of size 5 $\times$ 300 $\times$ 128 $\times$ 128.
>
> [1] Wan, Qiang, et al. "SeaFormer: Squeeze-enhanced Axial Transformer for Mobile Semantic Segmentation." The Eleventh International Conference on Learning Representations. 2022.

---

> ### Author Response · Authors · 2023-11-20
>
> Dear Reviewer MwLf:
>
> Thank you for your valuable comments on our paper. We have explained the aftermath of modality imbalance, way to improve efficiency and what does the dot product in MLP () means in Equation 1. It would be nice if you could let us know your opinions on our responses. Hopefully, our responses can solve your concerns. If you have additional questions, we would be happy to answer.
>
> Sincerely, Authors

---

### Official Review · Reviewer_qkmM · 2023-11-01

**Soundness:** 3 good
**Presentation:** 3 good
**Contribution:** 3 good
**Rating:** 6
**Confidence:** 4

**Summary:**

This article proposes a Bidirectional Interaction mechanism for the AVS task, which enhances the interaction between audio and visual. In terms of details, based on cross-attention, two modules, AGV and VGA, are used. An Audio Feature Reconstruction mechanism was also designed to address the problem of no supervision in the audio branch. With the support of these modules, this paper achieves state-of-the-art performance under AVS tasks. Ablation experiments demonstrate the effectiveness of the proposed module.

**Strengths:**

1. The paper is well written, the motivation is reasonable, and the reasons and practices for the design of each module are easy to understand.
2. The results in Table 5 surprise me. Increasing the number of decoder layers can actually lead to such a significant performance improvement.

**Weaknesses:**

1. Audio Feature Reconstruction seems to be very similar to the bidirectional generation module in AVSBG [1].
2. I can't see from Figure 5 which areas the attention mechanism obviously pays attention to.

[1] Dawei Hao, Yuxin Mao, Bowen He, Xiaodong Han, Yuchao Dai, and Yiran Zhong. Improving audio-visual segmentation with bidirectional generation. arXiv preprint arXiv:2308.08288, 2023.

**Questions:**

N/A

---

> ### Author Response · Authors · 2023-11-17
> **Responses to Reviewer qkmM**
>
> Thank you for your valuable comments. Below are our responses.
>
> **Q1: Audio Feature Reconstruction seems to be very similar to the bidirectional generation module in AVSBG [1].**
>
> **A1:** Our AFR module is quite different from the bidirectional generation module in AVSBG in 3 aspects: motivation, structure and constraint loss.
>
> **Motivation:** Our AFR aims to evade harmful data bias and also curtail audio information loss by reconstructing lost audio feature from AGV, while the bidirectional generation module in AVSBG just aims to build a strong correlation between visual and audio signals.
>
> **Structure:** The bidirectional generation module in AVSBG adopts a mask encoder-decoder structure to turn masked visual feature to reconstructed audio feature, while our AFR is simpler and more efficient in structure by using only two linear layers for projection.
>
> **Constraint loss:** AVSBG designs a consistency loss to minimize the distance between the norms of the original and reconstructed audio feature along the last dimension, while we adopt MSE loss to minimize the distance between the projected reconstructed audio feature and the projected multi-model feature.
>
> **Q2: I can't see from Figure 5 which areas the attention mechanism obviously pays attention to.**
>
> **A2:** The darker areas are visually very different from other lighter areas and highlight the attention regions of the network. You may regard the dark areas as where the attention pays attention to.

---

> ### Author Response · Authors · 2023-11-20
>
> Dear Reviewer qkmM:
>
> We thank so much for your valuable comments on our paper. We have clarified the differences between our AFR module and the bidirectional generation module in AVSBG and the areas in Figure 5 that the attention mechanism pays attention to. Could you let us know your thoughts on our responses? We hope our responses can address your concerns. We would be delighted to answer additional questions if any.
>
> Sincerely, Authors

---

### Official Review · Reviewer_7Z8k · 2023-11-02

**Soundness:** 3 good
**Presentation:** 3 good
**Contribution:** 2 fair
**Rating:** 6
**Confidence:** 4

**Summary:**

The paper describes an approach for audio-visual segmentation – where the goal is to segment sounding objects in a given video. The main focus of the paper is on improving audio-visual fusion by emphasizing on the audio modality. This is done through a bi-directional audio-visual decoder. An audio feature reconstruction is also used to further emphasize the audio. Experiments are done on the AVS Benchmark and results show that the proposed method obtains improvements of up to 1-4% in F-score and 2-5% in mIoU over prior work.

**Strengths:**

–  The direction of the paper is good. Most audio-visual work often end up focusing too much on the visual modality even if the task is acoustic in nature.

– The approach makes sense to my understanding and appears to be a simple but effective extension to improve audio-visual segmentation.

– Ablation studies have good coverage of different aspects of the method.

**Weaknesses:**

– While the emphasis on improving uses of audio cues is good, I am not sure if the claims around cross modal attention is entirely correct. There are multimodal works where both audio-visual attention is through both audio-video and video to audio. That is attentions of the forms Attn(Q_a, K_v, V_v) and Attn(Q_v, K_a, F_a) – so that both audio and visual features are obtained through cross attention. [1, 2] are just 2 examples, likely there are other papers. Not sure however if they have been used for AV segmentation task.

– Not clear about the claims of richness of information in audio – experimentally this is illustrated by showing that removing self-attention leads to better results. Can you discuss this in a bit more detail and clarify how this conclusion is reached?   It’s not clear that improved performance by removing self-attention can actually lead to this claim.

– I think defining L_dice, L_mask and L_afr clearly will add clarity to the the paper.

– How does the overall method behave when the sounding object is outside of the field of view or the sounding objects moves in and out of field of view.

– The AFR loss is essentially forcing feature similarity between F_AGV and F_audio (through the feature learning). Why would this reduce the type of bias mentioned in section 3.3 (single and mult-source etc.)

**Questions:**

Please respond to the questions in the weakness section.

---

> ### Author Response · Authors · 2023-11-17
> **Responses to Reviewer 7Z8k**
>
> Thank you for your valuable comments. Below are our responses.
>
> **Q1: Not sure however if cross model attention has been used for AV segmentation task.**
>
> **A1:** As far as we know, we are the first to use the cross model attention mechanism in AVS task, since most other AVS works only adopt either Attn(Q_a, K_v, V_v) or Attn(Q_v, K_a, V_a) instead of combining both. Though cross model attention has been used in multi-modal related tasks, the characteristics and motivation are always different from ours. Other works generally use this mechanism to build better correlation between two modalities and is often used only once, which makes the interaction insufficient. Prior works also did not consider from the modality imbalance aspect. While our design features both bidirectional and continuous modality interaction and our motivation is to enhance audio cues to mitigate modality imbalance issue.
>
> **Q2: Not clear about the claims of richness of information in audio. Removing self-attention leads to better results, discuss this in a bit more detail and clarify how this conclusion is reached.**
>
> **A2:** We claim in paper that the audio feature is not as information-rich as visual feature, so it may not need too much encoding operation, which means merely using cross attention is enough. The reason for the performance dropping when adding self-attention may be that adding self-attentions leads to overfitting. We demonstrate this by comparing the training and eval mask loss functions w/ and w/o self-attention and update the line chart in Figure 7 (b) in the appendix of our pdf file. We find that adding self-attention makes the mask loss drops lower while the test result is inferior to no self-attention case, which means overfitting.
>
> **Q3: Define L_dice, L_mask and L_afr more clearly.**
>
> **A3:** Thank you for your advice. We have clarified the definitions and highlighted them in orange color in the updated version. $L_{mask}$ = $L_{focal}$ + $L_{dice}$, $L_{afr}$ = MSE ($F_{afr}$, $F_{proj}$).
>
> **Q4: How does the overall method behave when the sounding object is outside of the field of view or the sounding objects move in and out of field of view.**
>
> **A4:** We visualize this case by comparing with AVSegFormer and AVSBench and the visual result is updated in Figure 7 (a) in our appendix. We find that present methods including ours still cannot well tackle this issue since in the last column the sounding ambulance car moves out of the view but all three methods falsely segment the motorcycle as the sounding object. This will be our future research direction. However, our method is still better than the other two since only our method correctly segments the ambulance car in view in the fourth column.
>
> **Q5: The AFR loss is essentially forcing feature similarity between F_AGV and F_audio (through the feature learning). Why would this reduce the type of bias mentioned in section 3.3 (single and multi-source etc.)**
>
> **A5:** We mitigate data bias from audio feature preservation aspect since we try to evade the harmful case where the network happens to “guess” the right target region even if the audio information has been lost or distorted. To achieve this, we design AFR module to reconstruct lost audio feature from AGV so that the "guess" case is curtailed. We have updated our pdf file and added visual comparison results with and without AFR to visualize its effect in Figure 8 in the appendix.

---

> > ### Comment · Reviewer_7Z8k · 2023-11-22
> > **Response after rebuttal**
> >
> > Thank you for providing detailed response including additional analyses and results. I will keep the score as is.  Some of the concerns could have been better addressed including those by other reviewers (e.g xbK3).
> > The AFR loss visualization (Fig 8) is a good addition but the explanation is still not convincing. I am not sure how data bias is coming into picture ? If there is a data problem and audio information is lost/distorted for some reason why would forcing the F_AGV to learn F_audio help ? Shouldn't it make things worse because the network is being forced to learn from mostly noisy information ? It seems to be simple way to raise the significance of audio driven feature learning ?

---

> > > ### Author Response · Authors · 2023-11-22
> > > **Responses to Reviewer 7z8k**
> > >
> > > Thank you for your valuable comments. Below are our responses. Hopefully our responses can solve your concerns.
> > >
> > > **Q1: I am not sure how data bias is coming into picture?**
> > >
> > > **A1:** [h] also adopt the same kind of reconstruction loss to reconstruct language features from multi-model features. The improvement brought by AFR is not due to data bias but because AFR guarantees the existence of audio cues to provide enough guidance to visual segmentation. We have updated more visual examples in Figure 8 in the appendix of our updated paper.
> > >
> > > **Q2: If there is a data problem and audio information is lost/distorted for some reason why would forcing the F_AGV to learn F_audio help ?**
> > >
> > > **A2:** We have added a visual result without audio cues in Figure 9 in appendix of out updated paper. Result show that without audio cues dramatically harms the AVS performance.
> > >
> > > **Q3:  Shouldn't it make things worse because the network is being forced to learn from mostly noisy information?**
> > >
> > > **A3:** You may refer to our visual results without audio enhancement in Figure 5, where lacking audio enhancement actually harms the final performance. At present, most work adopt pre-trained audio backbone (VGGish) to extract audio cues and then send to AVS networks. The extracted audio cues already have a relatively good quality.
> > >
> > > **Q4: It seems to be simple way to raise the significance of audio driven feature learning?**
> > >
> > > **A4:** Strengthening audio cues brings AVS performance gains. At present, most work adopt pre-trained audio backbone (VGGish) to extract audio cues before sending to AVS networks. Your proposal will be considered as our future research direction, thank you!

---

> ### Author Response · Authors · 2023-11-20
>
> Dear Reviewer 7Z8k:
>
> We deeply appreciate your valuable comments on our paper. We have provided corresponding responses to your questions. It would be great if you could let us know your thoughts on our responses. We hope our responses can address your concerns. We would be happy to answer additional questions if any.
>
> Sincerely, Authors

---

### Author Response · Authors · 2023-11-17
**Global response to all reviewers**

We thank all the reviewers for their insightful reviews. **We have updated our pdf file** with more figures and charts to demonstrate our statements in the appendix and also lighted some areas in the paper. Please check our updated file, thanks.

**We first summarize the strengths of our paper recognized by the reviewers.**
1. The direction of the paper to enhance audio cues is good. Most audio-visual work often end up focusing too much on the visual modality even if the task is acoustic in nature.
2. The approach makes sense to understanding and is a simple but effective extension to improve audio-visual segmentation.
3. The paper's originality is properly presented and mentioned, the motivation is reasonable and the connection of methods and performance is clearly presented.
4. The results present significant performance improvement.

**Next, we aim to address a common concern raised by reviewers (7Z8k, xbK3).**

1.The bidirectional framework (BAVD) and is not entirely novel.

Though the self-attention and cross-attention operations in our BAVD are not novel in multi-modal field, the novelty of our BAVD actually lies in its unique motivation and the simple but effective design to realize its motivation. The module based on a distinct motivation will offer a novel breakthrough point for AVS task.

The novelty of BAVD lies in motivation, and its simple but effective design that combines both bi-direction and continuity. The motivation of BAVD is to mitigate the modality imbalance issue that hasn’t been noticed by any other audio-visual works by strengthening audio cues. It is novel to break through from audio aspect because most audio-visual works often focus too much on the visual modality even if the task is acoustic in nature. BAVD also enables both bidirectional and continuous audio-visual interaction to prevent audio cues from losing during training, this is also different from other audio-visual works that either interact only once or unidirectionally.

---

### Meta-Review · Area_Chair_pSSS · 2023-12-06

**Metareview:**

This paper studies the audio-visual segmentation problem (AVS) which consists of segmenting an image into sounding objects (areas) given an audio query. The paper argues that current AVS methods suffer a "modality imbalance issue" and that "visual features take the dominance while audio signals easily fade away and are unable to provide enough guidance". To address this issue, the paper proposes a bidirectional audio-visual decoder (AVSAC) with cross-attention layers between audio and visual streams (rather than uni-directional or late fusion). In addition, an audio feature reconstruction (AFR) approach compensates data bias and aims to further preserve useful audio information. Experiments show that the proposed methods (when taken together) achieve better results than previous SOTA methods on a standard benchmark.

Strengths:
- The overall approach and motivation is intuitive and well described. Reviewers appreciate the general direction and motivation of "paying more attention to audio"
- The paper's originality is mostly presented and put into context
- The results show a noticeable performance improvement

Weaknesses
- The core idea of a bidirectional modality fusion framework (BAVD) is not particularly novel, although it may be used here for the first time on this particular problem -- the paper would be stronger if it had transferable and genuine novelty, rather than a "novelty of motivation, interaction mode and structure of our designs" (which feels a bit like an over-engineered specific solution)
- The reconstruction loss (AFR) is another core contribution, but presented in a lot less detail, not ablated in isolation and the explanations given are contradictory: "The improvement brought by AFR is not due to data bias" and "Our AFR aims to evade harmful data bias," and reviewers are not entirely convinced even after several rounds of discussions
- The AC finds the paper hard to read, as also noted by reviewers. Proofreading by a native speaker would help
- "Modality imbalance" is only introduced intuitively, the paper would be much stronger if there was e.g. an information-theoretic formulation or measurement of the contributions of the two modalities. "Data bias" is also not explained -- does it mean that there may only be one sounding region in an image?

**Justification For Why Not Higher Score:**

"Novelty of motivation, interaction mode and structure of our designs" is not a strong reason to accept this paper, no matter if the authors may have been inspired by the discussion of "modality imbalance" in [h] or not. While reviewers are mostly positive towards the paper, even after rebuttal, there is no strong excitement about the paper, and no strong reason to accept. Authors' style is quite vague and a bit hand-waving ("data bias", "modality imbalance"), which makes the paper hard to read, requiring a lot of context and intuition. A more precisely written paper may be easier to accept.

**Justification For Why Not Lower Score:**

n/a

---

### Decision · Program_Chairs · 2024-01-16

Reject